# Genotyping by Sequencing (GBS)-Based QTL Mapping for Bacterial Fruit Blotch (BFB) in Watermelon

**DOI:** 10.3390/genes13122250

**Published:** 2022-11-30

**Authors:** Sang-Min Yeo, Jeongeui Hong, Mohammad Rashed Hossain, Hee-Jeong Jung, Phillip Choe, Ill-Sup Nou

**Affiliations:** 1Department of Horticulture, Sunchon National University, Suncheon 57922, Republic of Korea; 2Department of Biological Sciences and Biotechnology, Chungbuk National University, Cheongju 28644, Republic of Korea; 3Department of Genetics and Plant Breeding, Bangladesh Agricultural University, Mymensingh 2202, Bangladesh; 4PPS Farming Corporation, Yongin 17096, Republic of Korea

**Keywords:** bacterial fruit blotch, quantitative trait loci, single nucleotide polymorphism, high-resolution melting

## Abstract

Watermelon (*Citrullus lanatus*), an economically important and nutritionally rich Cucurbitaceous crop grown worldwide, is severely affected by bacterial fruit blotch (BFB). Development of resistant cultivar is the most eco-friendly, cost-effective, and sustainable way to tackle this disease. This requires wider understanding of the genetics of resistance to BFB. In this study, we identified quantitative trait loci (QTLs) associated with BFB resistance in an F_2_ mapping population developed from BFB-resistant ‘PI 189225’ (*Citrullus amarus*) and -susceptible ‘SW 26’ (*C. lanatus*) genotypes based on the polymorphic markers identified by genotyping by sequencing (GSB). A linkage map covering a total genetic distance of 3377.1 cM was constructed. Two QTLs for BFB resistance, namely, *ClBFB10.1* and *ClBFB10.2*, both located on chromosome 10 explaining 18.84 and 15.41% of the phenotypic variations, respectively, were identified. Two SNP-based high-resolution melting (HRM) markers *WmBFB10.1* and *WmBFB10.2* having high positive correlation with resistance vs. susceptible alleles were developed. The efficacy of the markers was validated in another F_2_ population derived from SW34 × PI 189225. The highest phenotypic variation was found in the locus *ClBFB10.2*, which also contains three putative candidate genes for resistance to BFB. These findings will accelerate the development of BFB-resistant watermelon varieties via molecular breeding.

## 1. Introduction

Bacterial fruit blotch (BFB), caused by *Acidovorax citrulli* (formerly, *Acidovorax avenae* subsp. *citrulli*), is a devastating disease that causes severe yield losses in cucurbitaceous crops including watermelon (*Citrullus lanatus*) [1,2,3,4,5]. BFB was first reported in 1965, and an outbreak in watermelon was described in Florida in 1969 [1,6]. In South Korea, BFB in watermelon was first reported in 1990, and since then, it is a major concern for cultivation of watermelon [7].

Water-soaked lesions on cotyledon, leaf, and fruit are the typical symptoms of BFB; they are often small and irregular and spread throughout the leaf and rind of the fruit, eventually causing the plant to collapse [8]. This disease is a serious threat, especially since it affects not only the plant but also the fruit, greatly affecting the marketability of the crop. Furthermore, this seed-borne pathogen can survive in the seeds for more than 30 years, making it difficult to control in over-wintering and trans-boundary movement of the pathogen [9]. Cultural practices and fungicide application are currently used to manage BFB in watermelon, which often are not fully effective in the management of the disease [10]. Furthermore, fungicide applications raise production costs significantly, and repeated use may have a negative impact on the environment. The best alternative would be to use BFB-resistant cultivars, as host–plant resistance is the most effective control measure against bacterial plant diseases [11]. However, commercial watermelon cultivar with high BFB resistance is not available currently.

Several studies on cucurbit germplasm identified sources of host resistance for BFB [8,12,13,14,15,16]. Genetic control of the resistance to the disease is yet to be fully understood. Hopkins and Levi (2008) described the inheritance of resistance to BFB as complex interactions between multiple genes [17], while Islam et al. (2020) identified monogenic dominant control of BFB in melon [18]. Branham et al. (2019) described the foliar resistance to *A. citrulli* in *Citrullus amarus* (watermelon) to be complicated by low heritability, strong environmental influence and significant genotype-by-environment interactions [19]. Quantitative trait loci (QTL) for many diseases of watermelon, such as gummy stem blight (GSB), fusarium wilt, and papaya ringspot virus-watermelon strain (PRSV-W), etc., were identified recently [20,21,22,23,24,25,26]. For BFB, Branham et al. [13,19] identified six QTLs located on chromosomes 1, 2, 3 and 8 explaining 5% to 15% phenotypic variations for *A. citrulli*-induced percentage of affected leaf area in *C. amarus* line USVL246-FR2. Wu et al. [13,19], on the other hand, identified two additional QTLs on chromosomes 6 and 10 in a diverse *C. amarus*, *C. lanatus*, and *C. mucosospermus* population. In terms of molecular markers for BFB, only Islam et al. (2020) developed molecular markers linked to BFB in melon [18]. No such marker is available for BFB resistance in watermelon thus far.

In this study, we studied the resistance to BFB in the F_2_ population developed from resistant and susceptible watermelon lines, PI 189225 (*C. amarus*) and SW 26 (*C. lanatus*), respectively, based on genotyping by sequencing (GBS)-derived SNPs in a way to identify QTLs and to develop linked high-throughput molecular markers. This will enhance our understanding of the trait and will facilitate development of resistant varieties using marker-assisted selection in breeding programs.

## 2. Materials and Methods

### 2.1. Plant Materials

Resistant watermelon (*C. amarus*) genotype, PI 189225, was obtained from the United States National Plant Germplasm System (https://npgsweb.ars-grin.gov/gringlobal/search (accessed on 11 May 2019)), US Department of Agriculture (USDA), USA, and susceptible watermelon (*C. lanatus*) genotypes, SW 26 and SW 34 were obtained from Sunchon National University, Korea. The F_1_ populations were generated from SW 26 × PI 189225 and SW 34 × PI 189225 crosses, respectively, and the F_2_ population was developed by selfing the F_1_ plants of each population. A total of 214 and 62 F_2_ plants from the above two populations, respectively, were used for bioassay phenotyping. All watermelon plants were grown under long-day conditions (14 h light/10 h dark cycles) in a plant culture room at 24–28 °C and 60% relative humidity.

### 2.2. Pathogen Culture and Inoculation

*Acidovorax citrulli* bacterial isolate, 16-088 was collected from the National Institute of Horticultural and Herbal Science (NIHHS), Korea. The bacteria were cultured for 36 to 48 h at 28 °C on King’s B (KB) media supplemented with 100 µg/mL ampicillin until the formation of bacterial colonies. Bacterial suspensions were prepared by gently scraping off the bacterial colonies with a sterile L-shaped rubber spreader after inundating the culture plates with 5 mL of sterile, double-distilled water (DDW). The inoculum suspensions were diluted to a final concentration of 3 × 10^6^ colony-forming units (cfu) mL^−1^ for inoculation.

### 2.3. Phenotyping

Leaves of two-week-old plants of all 214 and 62 F_2_ plants of SW 26 × PI 189225 and SW 34 × PI 189225 populations, respectively, were inoculated by spraying the bacterial inoculum until runoff, and a secondary inoculation was performed 3 days after first inoculation to ensure effective inoculation of all plants [13,18,19]. Disease symptoms were assessed on inoculated leaves of parental, F_1_ and F_2_ populations at two weeks after inoculation (WAI) on a scale of 1 to 5, with 1, 2, 3, 4, and 5 indicating <1%, 1–10%, 11–30%, 31–50% and 51–100% affected leaf area, respectively (Figure 1). The ratio of inoculated to total-leaf area was multiplied by 100 to calculate the percentage of inoculated area. The percent disease index (PDI) was calculated using the following formula:$$\mathrm{PDI}=100\times \sum \frac{\mathrm{Sum}\;\mathrm{of}\;\mathrm{numerical}\;\mathrm{disease}\;\mathrm{rating}}{\;\mathrm{Number}\;\mathrm{of}\;\mathrm{plants}\;\mathrm{evaluated}\;\times \;\mathrm{maximum}\;\mathrm{disease}\;\mathrm{rating}\;\mathrm{score}\;}$$

The PDI ranges from 20 (when all leaves show scale 1) to 100 (when all leaves show scale 5 or plant is dead).

### 2.4. Extraction of Genomic DNA

Genomic DNA was extracted from young leaf samples of each watermelon plant using a DNeasy Plant Mini Kit (QIAGEN, Hilden, Germany) according to the manufacturer’s instructions. A ND-100 Micro-spectrophotometer was used to measure the concentration and purity of the extracted DNA (NanoDrop Technologies Inc., Wilmington, DE, USA).

### 2.5. Genotyping by Sequencing and QTL Mapping

Samples of 96 plants (two plants from each of the parents, SW 26 and PI 189225, and their F_1_, 45 resistant and 45 susceptible plants from the F_2_ generation derived from these parents) were used for genotyping by sequencing (GBS). Purification of the samples was performed using the QIAquick PCR Purification Kit (QIAGEN, Hilden, Germany) as per the manufacturer’s protocol. The HiseqX instrument (Illumina, San Diego, CA, USA) was used to perform GBS at the Macrogen Co. in Seoul, Korea. GBS data for PI 189225, SW 26, F_1_- and F_2_-generation were analyzed using a custom-designed method by DNACARE (Seoul, Korea) [26,27]. Picard and BWA-mem were used to mark duplicates and to align reads to the reference genome (97,103 watermelon genome, version 2), respectively [28,29]. Variants were called using the GATK Hablotypecaller-GVCF genotyper pipeline, which was then filtered using VCF tools (MQ 40 and FS > 60) [30]. The SNPs were filtered using a 40 percent genotype missing criterion and a P 0.001 segregation distortion criterion. The inclusive composite interval mapping feature of the QTL ICIMapping software version 4.2 was used to map QTLs. Within a 2-LOD interval of significant QTLs, the 97,103 genome v2 was used to find candidate genes (http://cucurbitgenomics.org/ (accessed on 14 September 2021)).

### 2.6. High-Resolution Melting Analysis

HRM curve analysis was used in conjunction with a 3-blocked and unlabeled oligonucleotide probe (HybProbe) specific to the SNP site to detect SNPs. The primers and probes were synthesized by Macrogen Co., (Seoul, South Korea). The HRM assay was performed using SYTO 9 green fluorescent nucleic acid stain (Invitrogen, Thermo Fisher Scientific, USA) on gDNA using a LightCycler 96 instrument (Roche, Mannheim, Germany). HRM curve analysis was performed using LightCycler 96 software version 1.1 (Roche, Mannheim, Germany) with a 0.02 positive/negative threshold level and 100 percent discrimination for both delta Tm and curve shape.

### 2.7. Statistical Data Analysis

The differences between means were determined using one-way ANOVA at 0.05 *p* value. A chi-square (χ^2^) test for goodness of fit was used to determine deviations of observed data from expected segregation ratios using the XLSTAT software. For all other analyses, the PRISM 6 software was used (ver. 6.01, GraphPad Inc., San Diego, CA, USA).

## 3. Results

### 3.1. Inheritance of Resistance to BFB in Watermelon

The percent disease index (PDI) of the resistant parent, PI 189225 (25.0), was significantly lower than that of the susceptible parent, SW 26 (68.3) (*n* = 10; *p* value < 0.001) (Figure 2). Their F_1_ (SW 26 × PI 189225) population also showed resistance response to BFB, as the PDI was only 35.0. The F_2_ (SW 26 × PI 189225) population showed a range of distributions of PDI starting from 20.0 to 100 with a population mean of 53.34. Population distribution for resistance to BFB was significantly deviated from a normal distribution according to the Shapiro–Wilk test for normality (*p* value = 0.001). The frequency distribution of PDI in the F_2_ population indicates a quantitative nature of inheritance of resistance to BFB in watermelon (Figure 2).

### 3.2. QTLs for Resistance to BFB

The genotyping by sequencing (GBS) of the population (96 plants including parental, F_1_ and F_2_ plants) derived from SW 26 × PI 189225 identified a total of 1,144,359 polymorphisms (SNPs and InDels) (Appendix A). Among these, 2461 were found be polymorphic between the resistant and susceptible genotypes. After filtering, 273 SNP markers were used to construct a genetic map encompassing a total length of 3377.1 cM (Figure 3 and Appendix A). The average length of linkage group was 307.0 cM with an average of 24.81 markers per linkage group and an average 15.31 cM interval between markers (Figure 3 and Appendix A).

A total of two QTLs, *ClBFB10.1* and *ClBFB10.2*, located on 12.5–16.5 and 515.5–521.5 cM of chromosome 10, respectively, were identified for resistance to BFB in the F_2_ population (Figure 4). The QTLs, *ClBFB10.1* and *ClBFB10.2*, explained 18.84% and 15.41% of phenotypic variations (R2) for resistance to BFB in the mapping population, with maximum LOD scores of 4.29 and 2.94, respectively (Table 1). The total number of genes in the 2-LOD interval for each QTL were *ClBFB10.1*: 15; *ClBFB10.2*: 18 (Appendix A). Since the additive effect of two QTLs was positive (>0), alleles conferring BFB resistance were contributed by resistant parent line ‘PI 189225’.

### 3.3. SNP Genotyping Using High-Resolution Melting (HRM) Assays

High-resolution melting (HRM) assays were developed for the SNPs, Chr10_6133795 and Chr10_32259129 closest to the peaks of *ClBFB10.1* and *ClBFB10.2*, respectively, and the developed markers are named as *WmBFB10.1* and *WmBFB10.2*, respectively (Table 2 and Appendix A). The HRM assay for SNP Chr10_32259129 (*ClBFB10.2*) showed a significant difference (*p* < 0.05) in PDI between the progenies homozygous for the susceptibility allele (SW 26, S/S) (49.92 ± 2.367) and for the resistant allele (PI 189225, R/R) (43.86 ± 3.1) in the F_2_ population (Figure 5A). Contrastingly, the HRM assay for the SNP Chr10_6133795 (*ClBFB10.1*) indicated a non-significant difference in the PDI between the progenies homozygous for the resistant allele (PI 189225, R/R) and the susceptibility allele (SW 26, S/S). The efficacy of the developed HRM markers was validated using another F_2_ population derived from SW 34 (♀) × PI 189225 (♂). In this population, similar trends of differences between PDI of the F_2_ progenies of the homozygous-resistant, homozygous-susceptible and heterozygous were observed (Figure 5B).

Similar trends of correlation between PDI were observed for homozygous-resistant, homozygous-susceptible, heterozygous and F_2_ progenies. Consistent with the above results, only *ClBFB10.2* showed a significant difference in PDI of the other F_2_ population derived from the resistant line, PI 189225 (*C. amarus*), and the susceptible line, SW 34 (*C. lanatus*).

### 3.4. Candidate Genes Identification

A total of 15 and 18 genes were identified within the regions of QTLs, *ClBFB10.1* and *ClBFB10.2*, respectively in the Cucurbit Genomics Database (http://cucurbitgenomics.org/ (accessed on 14 September 2021)) (Appendix A). QTL *ClBFB10.2* on chromosome 10 was responsible for the highest phenotypic variance and harbored several disease-resistance-related genes including one zinc-regulated protein gene (*ClaC10G202340*), one dehydration-responsive element binding transcription factor (*ClaC10G202360*), Trihelix transcription factor GT-2 (*ClaC10G202370*), etc. Notable biotic and abiotic stress-resistance-related genes within the QTL *ClBFB10.1* include calcium-dependent protein kinase 14 (*ClaC10G189800*), CTD small phosphatase-like protein 2 (*ClaC10G189830*), isoflavone reductase-like protein (*ClaC10G189890*), etc. The role of these genes in conferring resistance to BFB in watermelon needs to be investigated further.

## 4. Discussion

BFB caused by *Acidovorax citrulli* affects the leaves and fruits of watermelon, which greatly reduces the marketable yield. Sources of resistance to BFB have mostly been identified in citron melon (*Citrullus amarus*), a closely related species that crosses readily with watermelon and hence allows for introgression of resistant alleles into cultivated watermelon (*Citrullus lanatus* L.) [31,32].

This study focused on understanding the inheritance patterns, genetic determinants, and development of markers for resistance to BFB against the *A. citrulli* strain 16-088 in the population derived from contrastingly resistant parents SW 26 (S) × PI 189225 (R), belonging to *C. lanatus* and *C. amarus* species, respectively. The segregation ratio of F_2_ plants indicated that multiple genes were involved in conferring resistance to BFB in this population (Figure 1). This multi-genic control of the trait is in agreement with the previous findings, which suggested significant genotypic variations [8], polygenic control of the trait having low heritability, strong influence of genotype and environmental interactions [17], and significant genotype-by-year interactions [33]. In melon (*Cucumis melo*), however, Islam et al. (2020) [18] described monogenic dominant control of resistance to BFB by seedling bioassay screening of a segregating population of 491 F_2_ individuals developed from PI lines, PI 353814 (resistant) and PI 614596 (susceptible), against *A. citrulli* strain KACC18782 in greenhouse conditions.

Among the cucurbits, the genome of watermelon was first sequenced [34], with the genome of an East Asian cultivar, ‘97103’ [35] followed by an American watermelon ecotype, ‘Charleston Gray’ [13]. This paved the way for high-throughput genotyping and identification of QTLs associated with resistance to BFB in watermelon. Wu et al. [13,19] identified two QTLs for resistance to BFB by genome-wide association studies (GWAS) of a large collection of 1124 accessions of watermelon belonging to *C. amarus*, *C. lanatus*, and *C. mucosospermus*. In the same year, Branham et al. [13,19] identified six QTLs for resistance to BFB in a recombinant inbred population of 200 plants developed from USVL264-FR2 x USVL114, both belonging to *C. amarus* against *A. citrulli* strain AAC-001 in a thrice-replicated test using GBS. We identified two QTLs, *ClBFB10.1* and *ClBFB10.2* using GBS analysis of an F_2_ population derived from SW 26 x PI 189225 using the resistance source of PI 189225 (*C. amarus*) (Figure 3 and Table 1). The QTLs of Branham et al. [13,19] explained 5–15% of phenotypic variations, whereas QTLs of this study explained 15–18% of phenotypic variations (Table 1). All six QTLs of Branham et al. [13,19] were located on chromosomes 1, 2, 3 and 8 with two QTLs identified on each of the latter two chromosomes. Contrastingly, the QTLs of Wu et al. [13,19] were identified on chromosomes 6 and 10. Both of our QTLs were identified in chromosome 10 as well, of which the QTL *ClBFB10.2* (32.2–32.4 Mbp) was located in the adjacent region of chromosome 10 as that of the QTL (32.4–32.6 Mbp) of Wu et al. [13,19]. These differences could be due to the use of different populations in these two studies: ours on segregating the F_2_ population using PI 189225 (*C. amarus*) as a resistant source and Wu et al. [13,19] on a diverse population of 1124 accessions belonging to *C. amarus*, *C. lanatus*, and *C. mucosospermus*, possibly having multiple sources of resistance. Further fine mapping of the QTLs found on chromosome 10 of watermelon and their functional studies may lead to the causal gene.

We explored the genes within the regions of two identified QTLs, *ClBFB10.1* and *ClBFB10.2*. Within the QTL *ClBFB10.1* on chromosome 10 (1.18 Mbp), 15 genes were identified (Appendix A), of which three genes such as calcium-dependent protein kinase 14 (*ClaC10G189800*), CTD small phosphatase-like protein 2 (*ClaC10G189830*) and isoflavone reductase-like protein (*ClaC10G189890*) may have roles in plant defense response. Calcium-dependent protein kinases (CDPKs) are a large family of serine/threonine kinases that play an important role in plant defense [36] and phytohormone signaling pathways [37], and are responsible for the phosphorylation of a bZIP transcription factor FD, which is required for the formation of the florigen complex [38]. Glucose-6-phosphate/phosphate translocator 2 gene was found to be associated with verticillium wilt disease [39] and the C-terminal domain phosphatase-like 1 (CPL1) gene is known to repress stress-inducible gene expression in Arabidopsis [40] and enhance tolerance to heat stress in chrysanthemum [41]. Within the QTL *ClBFB10.2* region, a total of 18 genes were identified (Appendix A). Among these 18 genes, three genes including one zinc-regulated protein gene (*ClaC10G202340*), one dehydration responsive element binding transcription factor (*ClaC10G202360*), and Trihelix transcription factor GT-2 (*ClaC10G202370*) are thought to be linked to BFB resistance. Zn-deficient plants are more susceptible to disease in general. Many studies searching for a link between plant Zn status and disease severity have found that plants treated with Zn have a better response to fungi-caused disease [42]. A recent study reported that identification and characterization of DEAR1 (DREB and EAR motif protein 1; *At3g50260*), which encodes an EAR (ethylene response factor-associated amphiphilic repression) motif protein and acts as a transcriptional repressor of DREB1/CBF (dehydration-responsive element binding protein 1/C-repeat binding factor) protein [43]. Transgenic plants overexpressing DEAR1 showed increased resistance to pathogen infection [43,44]. Trihelix transcription factor GT2-like 1 (GTL1) was previously shown to play an important role in coordinating plant immunity [45,46].

One of the major drawbacks in the breeding of BFB-resistant cultivars has been the time-consuming and labor-intensive phenotyping process and the discrepancy in phenotyping results [13,19]. Development and application of molecular marker-assisted selection for molecular breeding can facilitate the development of BFB-resistant cultivar via reducing the time-consuming and labor-intensive steps in watermelon breeding. However, although QTLs related to BFB resistance have been reported in watermelon, molecular markers have not been developed yet. In melon, an InDel marker, *MB157-2-F/R*, based on a large insertion–deletion mutation between resistant and susceptible lines in TIR-NBS-LRR-encoding gene *MELO3C022157*, can genotype plants with high accuracy [18]. In this study, we developed and validated high-throughput HRM markers for MAS that will facilitate the selection of BFB resistance on chromosome 10. The marker WmBFB10.2 demonstrated a strong marker-trait association (Table 2; Figure 4), indicating its potential for genotyping resistant vs. susceptible plants. However, validation of the marker in a wide range of populations will be needed to use this marker across diverse genotypes of different watermelon species.

The QTLs and the developed marker, together with previously identified QTLs, will enhance our understanding of the inheritance of BFB resistance in watermelon and will be helpful in breeding programs aimed at developing watermelon cultivars with improved resistance to the disease.

## Figures and Tables

**Figure 1 genes-13-02250-f001:**
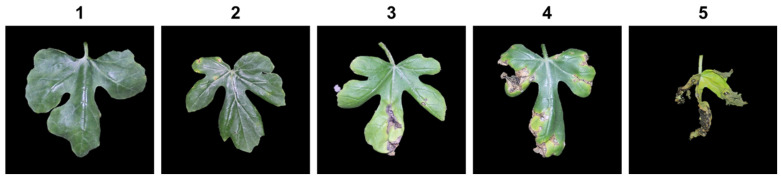
Scales showing the severity of disease symptoms on watermelon leaves infected with *Acidovorax citrulli*. Scores: **1** = <1%, **2** = 1–10%, **3** = 11–30%, **4** = 31–50%, **5** = 51–100% affected leaf area.

**Figure 2 genes-13-02250-f002:**
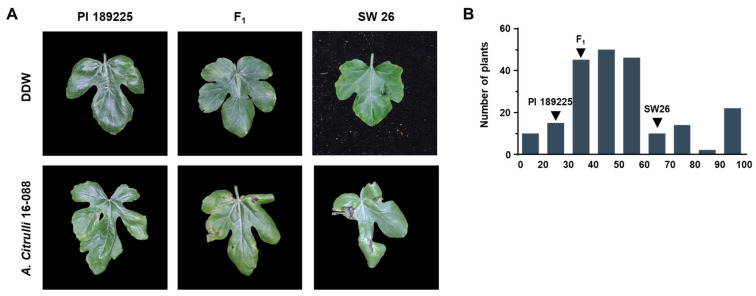
(**A**) Phenotype of the resistant (PI 189225), susceptible (SW 26), and their F_1_ (SW 26 × PI 189225) 14 days after inoculation (DAI) with DDW or *A. citrulli* strain 16-088. (**B**) Frequency distribution for percent disease index (PDI) at 14 DAI with strain 16-088 in the SW 26 × PI 189225 watermelon F_2_ population (*n* = 214). PI 189225, SW 26, and F_1_ population showed mean values (*n* = 10).

**Figure 3 genes-13-02250-f003:**
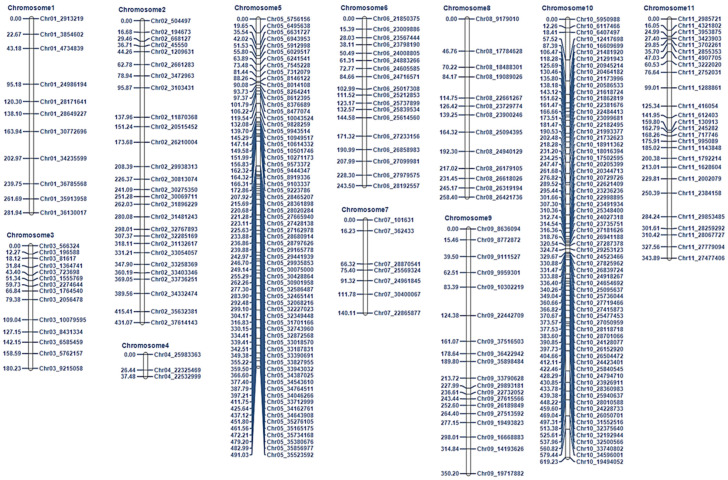
Genetic map of watermelon (SW 26 × PI 189225) segregating F_2_ mapping population comprising 275 genotyping by sequencing (GBS) derived SNP markers. The numbers on the left side represent the genetic distance from the top of each chromosome in cM. The names on the right side represent the name of the marker.

**Figure 4 genes-13-02250-f004:**
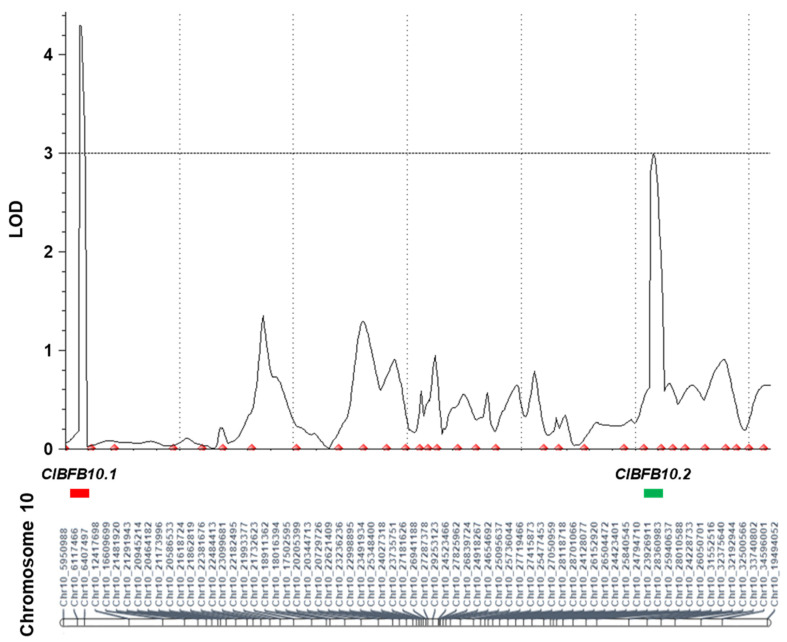
Quantitative trait loci (QTLs) associated with resistance to bacterial fruit blotch (BFB) on chromosome 10 of watermelon. QTLs are identified by composite interval mapping (CIM) analysis. LOD score at peaks are 4.292 and 2.939 for QTLs *ClBFB10.1* and *ClBFB10.2*, respectively. QTL is named according to: *Cl* (*Citrullus lanatus*)−BFB (trait name)–number (chromosome number).(unique identifier per chromosome).

**Figure 5 genes-13-02250-f005:**
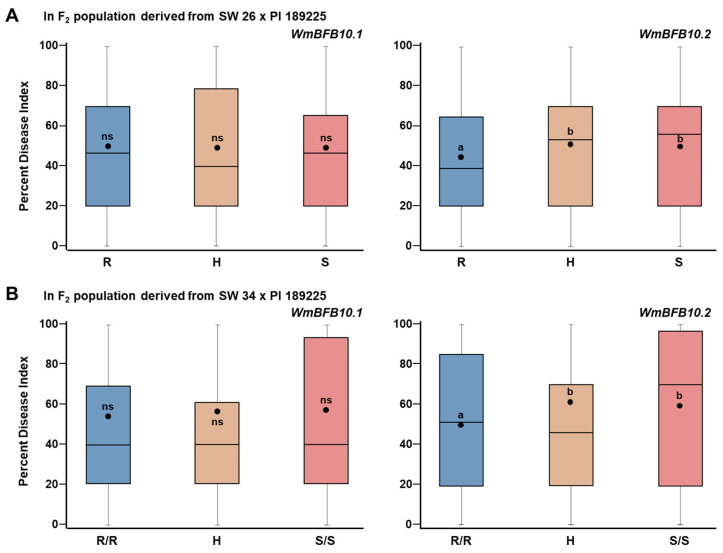
Box plots showing the differences in the percent disease index (PDI) of progenies of F_2_ population homozygous for resistant (R/R, PI 189225) allele, for susceptibility (S/S, SW 26) allele and for the heterozygous (H) allele as determined by the HRM assays for the SNPs, Chr10_6133795 and Chr10_32259129 in *ClBFB10.1* and *ClBFB10.2*, respectively. (**A**) F_2_ (SW 26 × PI 189225) (*n* = 214) and (**B**) F_2_ (SW 34 × PI 189225) (*n* = 62) population. Dots in the boxes represent the mean. (ns, no significance; *p* < 0.05, significant at 95% levels of probability in one-way ANOVA).

**Table 1 genes-13-02250-t001:** Details of the quantitative trait loci (QTLs) identified for resistance to bacterial fruit blotch (BFB) in watermelon.

QTL Name	Chromosome	Peak (cM)	LOD ^1^	Add ^2^	Dom ^3^	PVE ^4^	2–LOD Interval (cM) ^5^	Left Flanking Marker	Right Flanking Marker
*ClBFB10.1*	10	13	4.29	0.0708	−2.2148	18.84	12.5–16.5	6,117,466	6,407,497
*ClBFB10.2*	10	518	2.94	0.4174	1.9965	15.41	515.5–521.5	32,375,640	32,192,944

^1^LOD, logarithm of odds ratios at the position of the peak; ^2^ Add, additive effect of QTL; ^3^ Dom, dominance effect of QTL; ^4^ PVE, percentage of phenotypic variance explained by the QTL; ^5^ LOD, QTL interval on genetic map.

**Table 2 genes-13-02250-t002:** Primer sequences of SNPs associated with BFB resistance in watermelon.

SNP	QTL	Marker Name	Primer Sequence (5′–3′) *
Chr10_6133795	*ClBFB10.1*	*WmBFB10.1*	Forward: TTCAAACTGTAACAAAATAGAGTCAAA
			Reverse: AAATATTTGATTTTGAGTTCCTTCTCC
			Probe: TCCAGTCATCTCGTTTTACGCTTCAT
Chr10_32259129	*ClBFB10.2*	*WmBFB10.2*	Forward: AAGAGGTTTCGGATCAGGCCATGATT
			Reverse: GAACGCAAAATTAATTTCCAAACATATC
			Probe: TGATCGTTGAGAGCGAGGACGAAGT

* Underline indicates SNP.

## Data Availability

The raw sequence data from this study have been deposited in the publicly accessible National Center for Biotechnology Information (NCBI, https://www.ncbi.nlm.nih.gov/ (accessed on 17 October 2022)) database as PRJNA888469. The datasets supporting the conclusions of this article are included within the article and its additional files.

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
