# Peer review of "Genotyping by Sequencing (GBS)-Based QTL Mapping for Bacterial Fruit Blotch (BFB) in Watermelon"

_genes, 2022, doi:10.3390/genes13122250_

Round 1
Reviewer 1 Report
1. Bacterial Fruit Blotch (BFB) is important traits for watermelon production, the QTLs detection and molecular understand for this provide useful information for watermelon breeding.
2. In abstract section ,authors mentioned another two different F2 populations to confirmed the locus ClBFB10.2, but no any more information for these two population were mentioned. What is the purpose using two F2 population for this research, in the text body only SW 26 × PI 189225 was used?
3. For Materials section, information were not clearly provided in this part, how to investigated the performance? How to used two F2 generation for GBS? How to construct F2 and how many plants were investigated for each population were not cit. This section was so confused.
4.Line 77: There no figures for parental lines which could not understand materials clearly, and for Figure1, which generation were the individuals from?
5.Line 117-118:”Samples of 96 plants (two plants from each of the parents and the F1 generation, 45 resistant and 45 susceptible plants from the F2 generation) ........” which F2 generation?two F2 were used in present study?
6.Line 130, website or linkage for reference genome ?
Results
7.First , authors should let readers know how the two F2 population were constructed, which F2 population using for inheritance analysis or both?
8.Line 146-154: performance from SW26 crossing PI 189225 were investigated, how about SW34 crossing PI 189225 generation?
9.Figures of F2 by inoculated may help understand the performance.
10.How about the GBS data? A high density genetic map could help mapping QTLs, in this section, details for genetic map were not provided, and from Figure 3, chromosome 4 and 7 only 3 and 7 markers here, that was unacceptable for QTL detection. For chromosome 10, QTLs were detected because of the high density genetic map, so the results should be more accurate.
11.Line 203:how to identify “This HRM assay was validated using the F2 population derived from SW 34 (♀) × PI 189225 ...”?
12. The results section is confusing, many results lacking. Focus on the QTL mapping was more important, candidate gene identified should provide solid results, but this manuscript not. Two F2 generation were mentioned used, but few results here.
13. for discussion part:
Line 274-299: authors use a large part for discussion the candidate gene, but investigated the candidate gene should proved solid results, in this research, only one year performance was performed, and genetic map for some chromosome were not enough highly density to support the QTLs detection, I suggest focus on the genetic map construction and QTLs comparison with previous research.
TableS1 was not necessary.
Tables2 was a important results for results, suggested move to text.
All the figures were not clear, please check if fit the journal’s need.
Unfortunately the manuscript was poorly written in sciences. There are issues in data presentation and interpretation. Many details were also lacking. These issues greatly reduced the readability of the manuscript. Substantial revisions are needed in both language and science.
Author Response
Reviewer 1:
Bacterial Fruit Blotch (BFB) is important traits for watermelon production, the QTLs detection and molecular understand for this provide useful information for watermelon breeding.
Point 1: In abstract section, authors mentioned another two different F2 populations to confirmed the locus ClBFB10.2, but no any more information for these two population were mentioned. What is the purpose using two F2 population for this research, in the text body only SW 26 × PI 189225 was used?
Response 1: As part of our research with BFB breeding, we had developed several populations. In this study, we used two F2 populations derived from SW 26 x PI 189225 (population 1) and SW34 x PI 189225 (Population 2). Population 1 is used for Bioassay phenotyping, GBS, QTL mapping and to test the efficacy of the developed HRM markers. The population 2 was used as an additional validation population to test if the developed markers are also effective in selecting R v S lines in plant materials other than the original population 1 where those markers were developed from.
We understand that this was not clearly written in our original manuscript. We thank reviewer for pointing this out. We have now made corrections in abstract and results sections (shown below) to make this point clear to the readers.
In Abstract:
WmBFB10.1 and WmBFB10.2, two SNP-based high resolution melting (HRM) markers having high positive correlation with phenotypic variation were developed. The efficacy of the markers were validated in another F2 population derived from SW34 x PI 189225.
In results section:
The efficacy of the developed HRM markers were validated using another F2 population derived from SW 34 (♀) × PI 189225 (♂). In this population also, similar trends of differences between PDI of the F2 progenies of the homozygous-resistant, homozygous-susceptible and heterozygous were observed (Figure 5b).
In Figure:
In addition, the purpose of two populations is more clearly mentioned in the figure 5 now.
Please also note that all the corrections in this manuscript are made using ”Track Changes” option for ease in tracking.
Point 2: For Materials section, information were not clearly provided in this part, how to investigated the performance? How to used two F2 generation for GBS? How to construct F2 and how many plants were investigated for each population were not cit. This section was so confused.
Response 2:
We agree with these comments. Better clarification was necessary in manuscript.
The performance of the parental, F1 & F2 populations upon inoculation with A. citrulli is evaluated using a score (1-5) and percent disease index (PDI) was calculated using an equation. The details are now given in the ‘pathogen culture and inoculation’; and ‘phenotyping ‘part of ‘materials and methods’ section.
As for the GBS, we only used one F2 population (SW26 x PI189225) for GBS.
The F2 were constructed by selfing the F1 plants and 214 and 62 plants for SW26 x PI189225 and SW34 x PI189225 populations, respectively were used. This is now clarified in the ‘Materials and Methods’ section by including the following statement.
“and the F2 population were developed by selfing the F1 plants of each population. A total of 214 and 62 F2 plants from this above two populations, respectively were used for bioassay phenotyping.”
We believe this section is clear now.
Point 3: Line 77: There no figures for parental lines which could not understand materials clearly, and for Figure1, which generation were the individuals from?
Response 3: In lines 77, In Figure 1, leaves of two-week old plants (2–3 leaf stage) of were inoculated by spraying. Disease symptoms were assessed on inoculated leaves (4-5 leaf stage) after two weeks after inoculation. In figure 1, only the data of population 1 (SW 26 × PI 189225) is mentioned, which is now mentioned in the figure caption.
This type of bioassay is done for population 2 as well (SW 34 × PI 189225), but as that population were only used for validation of HRM markers we did not show their figures.
Point 4: Line 117-118:”Samples of 96 plants (two plants from each of the parents and the F1 generation, 45 resistant and 45 susceptible plants from the F2 generation) ........” which F2 generation?two F2 were used in present study?
Response 4: We thank reviewer for this comment. Indeed, it needed to be clarified.
We used the SW 26 × PI 189225 population and modified the statement to make it clear.
Point 5: Line 130, website or linkage for reference genome ?
Response 5: We have added this “(http://cucurbitgenomics.org/)” in line 130.
Point 6: First, authors should let readers know how the two F2 population were constructed, which F2 population using for inheritance analysis or both?
Response 6: The construction of F2 population is now clearly described.
We have revised this in the current version of the manuscript (please see ‘plant materials’ part of ‘materials and methods. section).
Point 7: Line 146-154: performance from SW26 crossing PI 189225 were investigated, how about SW34 crossing PI 189225 generation?
Response 7: Thank you very much for your comments. We have added this “In addition, SW 34 showed susceptible response to BFB as the PDI was 71.3, and PDI of F1 (SW 34 × PI 189225) population was 53.0”.
Point 8: Figures of F2 by inoculated may help understand the performance.
Response 8: Mentioning the figures of all 214 & 62 F2 plants will be too much. That’s why we decided not to overload the data. However, the representative symptoms of scores 1-5, based on which we have phenotype our F2 populations are clearly shown in Figure 1. We believe this is sufficient. We think, the reviewer will understand our point.
Point 9: How about the GBS data? A high density genetic map could help mapping QTLs, in this section, details for genetic map were not provided, and from Figure 3, chromosome 4 and 7 only 3 and 7 markers here, that was unacceptable for QTL detection. For chromosome 10, QTLs were detected because of the high density genetic map, so the results should be more accurate.
Response 9:
GBS analysis of 96 plants (two plants from each of the parents, SW 26 and PI 189225, their F1, 45 resistant and 45 susceptible plants from the F2 generation derived from this parents) identified a total of 1,144,359 polymorphisms (SNPs and InDels), of which , 2,461 were found be polymorphic between the resistant and susceptible genotypes. We have further filtered those polymorphisms based on robust filtration criteria (> 40% genetic results loss, and Segregation distortion; P>0.001) which finally yielded 273 markers. We have used such strict filtration criteria to avoid false positive QTLs. We believe that our linkage map is robust and the detected QTLs are solid. The developed markers from these QTL peaks also proves the robustness our QTLs as those markers were effective in differentiating R vs S loci not only in the original F2 population (SW 26 × PI 189225), but also in another test population (SW 34 × PI 189225).
The reason for being too strict with the filtration criteria is that our lab is working on developing resistant lines. And we want robust QTLs, linked markers and use those for marker assisted selection. For this time we want to stick with these results.
However, we agree that a high density map may give few more QTLs.
Point 10: The results section is confusing, many results lacking. Focus on the QTL mapping was more important, candidate gene identified should provide solid results, but this manuscript not. Two F2 generation were mentioned used, but few results here.
Response 10: Our main target was to identify QTLs for the resistance sources available in our lab and develop markers so that these can be used for developing resistant lines via marker assisted breeding. Candidate genes within the QTL regions are only mentioned as supporting to build a ground for future research. The next research can focus on confirming the candidature for resistance to BFB among these listed genes. So, in this article, candidate genes are only supporting information.
Regarding the two F2 populaiton, we believe their purpose and use are now clearly mentioned in the manuscript (as indicated in previous comments). This was truly a good suggestion, otherwise any reader would be confused about two populations. We that reviewer for pointing it out. It certainly improved our manuscript.
Point 11: Line 274-299: authors use a large part for discussion the candidate gene, but investigated the candidate gene should proved solid results, in this research, only one year performance was performed, and genetic map for some chromosome were not enough highly density to support the QTLs detection, I suggest focus on the genetic map construction and QTLs comparison with previous research.
Response 11: Thank you very much for your comments. There are currently few studies of BFB QTL in watermelon. Our findings suggest that chromosome 10 is related, which is similar to the results of Wu et al., 2019, Plant Biotechnology Journal. We reported QTL related to BFB and identified candidate genes, and we intend to conduct additional research on these candidate genes in the future.
Point 12: Unfortunately the manuscript was poorly written in sciences. There are issues in data presentation and interpretation. Many details were also lacking. These issues greatly reduced the readability of the manuscript. Substantial revisions are needed in both language and science.
Response 12: The manuscript is now thoroughly revised to improve the language (marked as track change throughout the manuscript). And addressing the reviewers comments certainly helped to improve the science of this manuscript as well. We believe this version is quite improved now. We thank reviewer for all valuable comments. This comments really helpful to find and address the pitfalls of this manuscript.
Reviewer 2 Report
The current manuscript, “Genotyping by sequencing (GBS) based QTL Mapping for Bacterial Fruit Blotch (BFB) in Watermelon” is an interesting work related to the identification of markers for BFB in Watermelon. However, the method methodologies were not enough. I suggest the author add some more experiments, add expression results of genes, and seriously revise the discussion section.
1.Line, 168 F2, correct it to F2
2.Figure 3 is not clear; please increase the resolution
3.Figures were referenced more than once, and please remove repeats.
4.Is this journal format? Why are the figures not labelled with letters?
5.If you could do an expression analysis of these genes by RTq-PCR, it could help to find the candidate gene.
6.Please explain the discussion in more detail and use more literature; half of the discussion references were only from two publications.
7.The gene names should be italic
8.In the discussion, an attempt should be made to discuss the biological role of the markers and candidate gene(s). Also, add more comparative results explaining and supporting the physiological and genetic regulatory pathways.
Author Response
Point 1:
- Line, 168 F2, correct it to F2
- Figure 3 is not clear; please increase the resolution
- Figures were referenced more than once, and please remove repeats.
- Is this journal format? Why are the figures not labelled with letters?
- The gene names should be italic
Response 1: We have revised this in the current version of the manuscript.
- A better version is being included.
- Unnecessary repetition is removed.
- The figures are now labelled.
- Gene names are now italicised.
Point 2: If you could do an expression analysis of these genes by RTq-PCR, it could help to find the candidate gene.
Response 2: Our main target was to identify QTLs for the resistance sources available in our lab and develop markers so that these can be used for developing resistant lines via marker assisted breeding. Candidate genes within the QTL regions are only mentioned as supporting to build a ground for future research. The next research can focus on confirming the candidature for resistance to BFB among these listed genes. So, in this article, candidate genes are supporting information. We are conducting follow-up studies on these candidate genes. But this may need some time. And, we are afraid we won’t be able to include those in this manuscript.
Point 3: Please explain the discussion in more detail and use more literature; half of the discussion references were only from two publications. In the discussion, an attempt should be made to discuss the biological role of the markers and candidate gene(s). Also, add more comparative results explaining and supporting the physiological and genetic regulatory pathways.
Responses to point 3: This is an obvious comment. Wider discussion would certainly enlighten the potential reader about the topics. We have tried our best to improve the discussion part. Actually researches on watermelon BFB resistances is scarce. So the comparative discussion was limited for this manuscript. However, we focused on the genetics of the trait, QTLs identified so far, potential candidate genes and their putative roles, our development markers for BFB & their practical implications within the wider context of mapping and molecular breeding of cucurbits. We believe this would be sufficient for BFB genetics in melon, a less studied disease in cucurbits.
Round 2
Reviewer 1 Report
I am happy to see significant revisions made to address issues in my previous review.
Methods of QTL mapping were also need presented in the text.
Author Response
Thank you very much for your comments. We described the methods of QTL mapping on lines 126-133.